# A Study of Films Incorporating Magnetite Nanoparticles as Susceptors for Induction Welding of Carbon Fiber Reinforced Thermoplastic

**DOI:** 10.3390/ma13020318

**Published:** 2020-01-10

**Authors:** Inseok Baek, Seoksoon Lee

**Affiliations:** Department of Mechanical Engineering and Engineering Research Institute, Gyeongsang National University, 501, Jinju-daero, Jinfu-si, Gyeongsangnam-do 52828, Korea; ttnever87@gnu.ac.kr

**Keywords:** induction welding, thermoplastic, susceptor, magnetite (Fe3O4), polyamide 6 (PA6), lap shear strength (LSS)

## Abstract

Induction welding is a fast, clean, noncontact process that often uses a metal-mesh susceptor to facilitate localized controlled heating; however, the metal mesh presents various problems. In this study, the induction heating behavior of a 450 μm thick thin-film susceptor, fabricated by mixing magnetite (Fe${}_{3}$O${}_{4}$) nanoparticles (NPs) and PA6/carbon fiber (CF) (30%) thermoplastic resin, was examined with respect to the weight ratio of Fe${}_{3}$O${}_{4}$ (50, 67, 75, and 80 wt%). The useful induction heating behavior of the 75 wt% Fe${}_{3}$O${}_{4}$ susceptor suggested its suitability for additional heat treatment experiments, carried out at 3.4 kW at a frequency of 100 kHz. This susceptor attained the same maximum temperature during 10 cycles of repeated induction heating and cooling. It was then used to weld two thermoplastic composites, with 60 s of induction heating followed by 120 s of simultaneous cooling and pressing. The resulting welded joints had lap shear strength values of 36.8, 34.0, and 36.4 MPa under tensile test loads of 884, 817, and 874 N, respectively. Scanning electron microscopy images confirmed a uniform weld quality. Thus, the proposed manufacturing method, involving the incorporation of Fe${}_{3}$O${}_{4}$ NPs into thermoplastic resin, should help expand the range of applications for thermoplastic composites.

## 1. Introduction

Many airliner structures, as well as automobile frames are made from composites, because the incorporation of composites offers the advantages of the shorter processing times and reduced weight of the fabricated structure. Thermosetting composites have commonly been used for these applications, but these composites are increasingly being replaced by thermoplastics. Thermosetting resin is easy to work as it is maintained in a liquid state at room temperature. A laminator can easily remove air during the manufacturing process and quickly produce products via vacuum or positive pressure pumps. The major advantages of thermosetting resin are the ease of manufacture and low raw material costs. However, thermosetting resins cannot be restored to their original condition by catalysis. That is, once the thermosetting composite is formed, it cannot be molded or deformed again. For this reason, recycling of thermosetting composites is very difficult. Since thermoplastic resin is solid at room temperature, it is difficult to impregnate the reinforcing fibers. The resin must be heated to the melting point; pressure is required to impregnate the fibers; and the composite must be cooled under this pressure. However, since reheating is possible, the thermoplastic composite material can be modified and remolded. Thermoplastics offer equivalent performance to thermosetting composites and can be recycled via heating. Thermoplastics are superior in toughness and environmental resistance and are superior to most thermosetting resins in terms of their short processing time, incombustibility, and long lifecycle. Additionally, thermoplastics can be produced quickly at low cost [1,2,3,4].

However, thermoplastic resins can be deformed only to a limited degree: the thermoplastic parts produced today have simple shapes, and complex shapes are needed to combine parts. Bonding is an important step in the process of manufacturing thermoplastic composites, which may weaken because bonding can cause irregularities in the structure. Conventional joining methods for metals and thermosets (mechanical fastening and adhesive bonding) can be used, but are not suitable for thermoplastics. Mechanical fastening has many disadvantages, including a concentration of stress in the material, peeling during drilling, a difference in thermal expansion between the fastener and the composite, water penetration into the joint, galvanic corrosion, weight gain, and extensive labor and time requirements. Adhesive bonding is superior to mechanical fastening because it avoids stress concentration, but difficulties are still encountered when it is applied to thermoplastics. Extensive surface treatments are required, but these are generally difficult to control in industrial environments, and adhesives (usually epoxy) have a long curing cycle [5,6,7].

Welding processes attempt to bond materials in such a way that allows them to retain their mechanical performance. Resistance, ultrasonic, and induction welding are widely used to bond thermoplastic composites [8,9,10,11]. In particular, induction welding is a fast, clean, non-contact process that can be applied to complex shapes. Unlike other types of welding, induction welding requires a susceptor: heating is controlled with a susceptor placed between two adherends. An eddy current generated by the magnetic field from an induction coil heats the susceptor, which in turn melts the two adherends. The molten adherends become welded when solidified under pressure. Most susceptors are made from iron oxide, nickel, or stainless steel. A stainless steel mesh is commonly used as the susceptor in thermoplastic composite induction welding. However, adhesion failure with the resin, uneven heating, an increase in weight, and residual stress continue to undermine the quality of thermoplastic welds [12]. Some researchers have attempted to solve these problems by incorporating micron sized (or smaller) heating particles into the resin [13]. Farahani [12] studied the effects of silver nanoparticles (NPs) on induction heating. Kwon [14] examined induction heating behavior according to the sizes of iron, iron oxide, and nickel NPs. Other researchers have investigated the feasibility of fabricating a thin-film susceptor from a mix of powdered ferromagnetic material and thermoplastic resin, and several studies have presented new susceptor configurations [15].

Various substances are used in induction heating, including magnetite (Fe${}_{3}$O${}_{4}$). Kwon [14] demonstrated that Fe${}_{3}$O${}_{4}$ heats well during induction heating of Fe, Fe${}_{3}$O${}_{4}$, and Ni. However, no studies have used Fe${}_{3}$O${}_{4}$ directly as the susceptor. In this study, a susceptor was prepared by mixing polyamide 6 (PA6) resin and Fe${}_{3}$O${}_{4}$. Induction welding was performed on a thermoplastic fiber reinforced plastic consisting of PA6/carbon fiber (CF, 30%). The induction heating behavior was tested by adjusting the mixing ratio of PA6 and Fe${}_{3}$O${}_{4}$, and a good susceptor was identified based on the mixing ratio at which a high temperature occurred. Since thermoplastics can be reheated, 10 reheating tests were performed to identify whether there was a change in the induction heating characteristics. Induction welding was performed using a 10 kW heating device. Specimens were fabricated using the single lap joint method, and a tensile test was performed to determine the weldability and mechanical performance by calculating the lap shear strength (LSS). This study is intended to verify the induction welding performance of thermoplastic composites by fabricating new susceptors.

## 2. Detail of the Experiment

### 2.1. Fiber Reinforced Plastic for Induction Welding

Thermoplastic PA6 (CF 30%) was used as the fiber reinforced plastic for induction welding. The thermoplastics most often used in aviation and automotive are polyetheretherketone (PEEK) and polyphenylenesulphide (PPS), but PA6 was chosen in this study. As shown in Table 1, PEEK has a high working temperature of $250{\;}^{\circ }C$; PPS has a low working temperature, of only $170{\;}^{\circ }C$, but it is difficult to obtain its raw materials as a powder. PA6 was chosen in this study because it can be molded at $180{\;}^{\circ }C$ and comes in powder form. Powder is needed for the raw material because it must be mixed with nano sized Fe${}_{3}$O${}_{4}$. Furthermore, fiber reinforced plastics have higher forming temperatures. PA6 (30% carbon fiber reinforced (CFR)) has a working temperature of $200{\;}^{\circ }C$, higher than that of PPS (20% CFR); therefore, PA6 thermoplastic was used in this study.

### 2.2. Fabrication of the Susceptor

The susceptor was made from a thermoplastic PA6 resin powder (average particle size: 50 μm) and Fe${}_{3}$O${}_{4}$ powder (average particle size: 200 nm), in which the weight percentage of Fe${}_{3}$O${}_{4}$ was varied. Figure 1 shows the susceptor fabrication process. First, PA6 and Fe${}_{3}$O${}_{4}$ powders were mixed thoroughly at a specific ratio. The PA6 was melted by heating the powder mixture to $230{\;}^{\circ }C$ using a hot plate. Upon cooling, a 450 μm thick PA6/Fe${}_{3}$O${}_{4}$ susceptor was produced. If you want to watch fabrication process, the fabrication of susceptor can be viewd in Appendix A. Increasing the weight ratio of Fe${}_{3}$O${}_{4}$ in the mixture increased the density of Fe${}_{3}$O${}_{4}$ in the susceptor and, potentially, the induction heating effect. The weight percentage of the Fe${}_{3}$O${}_{4}$ additive was varied in an attempt to identify the optimal weight percentage. The Fe${}_{3}$O${}_{4}$ weight percentages examined were 50, 67, 75, and 80 (Table 2).

### 2.3. Characteristics of Induction Heating Behavior

The induction heating behavior was examined using an infrared camera (Ti450PRO, FLUKE, Everett, WA, USA, −$10{\;}^{\circ }C$–$1500{\;}^{\circ }C$). The induction heating test was performed for 90 s, with a heating time of 30 s and a cooling time of 60 s at room temperature. The heating coil was a multi-turn type with an inner diameter of 15 mm, and the susceptor was heated at the center of the coil (Figure 2b). In this coil, an AC current flowed and generated an eddy current in the susceptor, thereby heating it. Because the workpiece could not be pressed while inside the coil, the coil was removed immediately after induction heating, and the workpiece was then pressed. The degree of cooling generated during the process of removing the coil was measured. We then measured the temperature and time required for the susceptor to melt the adherend during the induction heating process. The optimal weight percentage of Fe${}_{3}$O${}_{4}$ for induction welding was selected from among susceptor samples containing 50, 67, 75, and 80 wt% Fe${}_{3}$O${}_{4}$. The induction heating characteristics of the susceptor were obtained through 10 heating cycles of the chosen susceptor. This susceptor was then used in induction welding experiments to determine its ability to weld together two thermoplastic composites.

### 2.4. Induction Welding of Thermoplastic Composites

Induction welding of the thermoplastic composite was performed using the optimal as fabricated PA6/Fe${}_{3}$O${}_{4}$ thin-film susceptor. Figure 3 shows the induction welding process. Two pieces of the composite placed on either side of the susceptor were centered inside an induction heating device (HF-10K model, T.I.H., Korea, power consumption: 10 kW; frequency range: 100–400 kHz;). The susceptor and composite adherends were heated for 60 s, followed by simultaneous cooling and pressing of the composite joint for 120 s. A load of 5 kg was used to apply pressure to create the welded joint between the composites. In particular, the induction heating of the susceptor should be at least $300{\;}^{\circ }C$. When the susceptor is more than $300{\;}^{\circ }C$, its heat transfer was enough to be welded to the susceptor and the joint surface. The welded specimens were produced following the guidelines outlined in ASTM Standard D5868; Figure 4a shows the drawing of a specimen. The power output for the experiments was 3.4 kW at a frequency of 100 kHz. The output current was 45 A for the induction welding experiments. An AC current was applied to the induction heating coil [16,17,18].

### 2.5. Tensile Test

Mechanical testing of the welded thermoplastic composites was carried out using a servo tester (5883; Instron, USA) in accordance with ASTM Standard D5868 to measure LSS. The LSS of the welded specimen was measured using a 5 kN load cell, with a crosshead speed of 1 mm/min at room temperature. The LSS value was calculated based on the maximum force achieved before the specimen was destroyed, as follows:(1)$$\tau =\frac{F_{max}}{L\times b}\times \frac{N}{{\mathrm{mm}}^{2}}$$
where τ is the LSS (in N/mm${}^{2}$), *L* is the length of the overlap (mm), *b* is the width of the overlap (mm), and *F*${}_{max}$ is the maximum tensile force (*N*). In this study, *L* = 6 mm and *b* = 4 mm. The cross-sections of the specimens were observed using field-emission scanning electron microscopy (FE-SEM; MIRA3 LM; Tescan, Czech Republic) to assess the weld quality.

## 3. Results and Discussion

### 3.1. Induction Heating Behavior of the Susceptor

SEM images were taken before the induction heating experiment of the as-fabricated susceptor. Figure 5 shows the obtained susceptor; white areas are Fe${}_{3}$O${}_{4}$, and black areas correspond to PA6 resin. Fe${}_{3}$O${}_{4}$ appeared to be distributed uniformly throughout the resin. Thus, we expected uniform heating from the as-fabricated susceptor during induction.

In the induction heating experiments, the temperatures of four susceptors varying in Fe${}_{3}$O${}_{4}$ weight percentage (50, 67, 75, 80 wt%) were measured. All susceptors reached $200{\;}^{\circ }C$ or higher within 10 s (Figure 6a). The maximum temperatures of the four susceptors were 340, 250, 330, and $230{\;}^{\circ }C$, respectively. A dip feature appeared in the graph of the 50 wt% Fe${}_{3}$O${}_{4}$ sample; this was an error originating from the infrared thermal camera. If we considered the remaining three samples, the highest temperature measured by the thermal imaging camera was achieved by the 75 wt% Fe${}_{3}$O${}_{4}$ susceptor, so this susceptor was used in the repeated induction heating tests.

Ten iterations of induction heating were carried out with the 75 wt% Fe${}_{3}$O${}_{4}$ susceptor to determine whether degradation of the susceptor occurred with repeated heating. The first heating iteration had the fastest heating rate (Figure 6b). The second and all subsequent heating iterations had a constant heating rate, with similar maximum temperatures. The results were repeatable, so reheating did not appear to cause any deterioration of the susceptor sample.

### 3.2. Mechanical Performance of the Weld

Tensile tests were performed to measure the LSS performance of the welds created using the as-fabricated PA6/75 wt% Fe${}_{3}$O${}_{4}$ susceptor. The sample was evaluated on the basis of the acceptable LSS range (22.2–31.3 MPa) established by Farahani [12]. Table 3 presents the tensile test results for three welded composite specimens. As can be seen, all of the values were larger than the reference LSS range [12].

Figure 7 shows the adhesion failure of the weld specimens after the tensile test. Fracture occurred at some distance from the center of the weld in all specimens, so the joint itself held under the applied load.

### 3.3. Observation of the Welded Joint

The welded area was cut, and SEM images were taken of the cut section, as shown in Figure 8. Using magnifications from 200 to 30,000 times, it was confirmed that high density Fe${}_{3}$O${}_{4}$ appeared white and was uniformly mixed into the low density PA6 resin, which appeared black. No pores or cracks were observed in the adhesion area. In Figure 8b, the bonded cross-section is shown. Unlike the central plane, it can be seen that the resin was not distributed at the corners. It was apparent that the resin did not spread uniformly to the edges. Except for the corners, the resin was melted by induction heating and then cooled. After induction heating, the molten resin was spread by the load, which was applied simultaneously as the cooling. These images confirmed that induction uniformly heated the sample, the surface of the adherend melted properly, and a boundary layer appeared.

## 4. Conclusions

In this study, we fabricated thin-film susceptors for induction welding from a thermoplastic composite made of PA6 resin and Fe${}_{3}$O${}_{4}$ NP additive. The selected weight percentage for susceptor fabrication was identified as 75 wt% Fe${}_{3}$O${}_{4}$. Induction welding was performed using this susceptor, and the weld strength and quality were verified by tensile tests and SEM analysis of the weld joint cross-section. The results are summarized below.

Susceptor samples with varying amounts of Fe${}_{3}$O${}_{4}$ (50, 67, 75, and 80 wt%) all heated to $200{\;}^{\circ }C$ within 10 s. The maximum temperature for induction heating/welding applications was attained for the susceptor sample containing 75 wt% Fe${}_{3}$O${}_{4}$. Notably, the maximum heating temperature did not increase as the weight ratio increased. The average thickness of the susceptor used in the experiment was 450 ± 20 μm.The 75 wt% Fe${}_{3}$O${}_{4}$ susceptor sample was used in induction heating experiments, in which the sample underwent 10 cycles of heating for 30 s, followed by cooling for 30 s, to ascertain whether the heating performance of the susceptor remained similar. The heating rate was faster during the first iteration of induction heating. In subsequent iterations, the same heating behavior was observed, with the same maximum temperature. Thus, reheating did not affect the performance of the thermoplastic resin, as expected. As such, the performance of the as-fabricated susceptor with Fe${}_{3}$O${}_{4}$ additive was better than that of previous susceptors.When only the susceptor was induction heated for 30 s, it was heated to $340{\;}^{\circ }C$. Because of the temperature being transferred to the joint surface during induction welding, unlike the previous reheating experiment, 60 s of heating was applied in induction welding. In addition, a time of 30 s or more was required for the resins of the susceptor and the joint surface to melt. Induction welding was performed by melting the susceptor and the joint surface.Previous studies of thermoplastic composite joint performance reported high quality welds having LSS values of 22.2–31.3 MPa [15]. The tensile test results of this study, using the as-fabricated susceptor with 75 wt% Fe${}_{3}$O${}_{4}$, produced welded thermoplastic joints in three specimens with LSS values of 36.8, 34.0, and 36.4 MPa under loads of 884, 817, and 874 N, respectively.Cross-sectional SEM images of the weld joint with the 75 wt% Fe${}_{3}$O${}_{4}$ susceptor revealed no defects or breakages; additionally, Fe${}_{3}$O${}_{4}$ appeared to be distributed uniformly throughout the PA6 resin. Looking at Figure 8c-1, in the center, the part that looks like horizontal line is the welded joint. The enlarged picture shows that the Fe${}_{3}$O${}_{4}$ was properly mixed with the resin. Some of the previous studies showed that nanoparticles enhanced the mechanical properties [19]. The susceptor in this study differed from the previously studied susceptors by using the same resin as the adherend. Therefore, it is easy to apply to thermoplastic products.Previous research results showed that induction welding was possible from 60 Hz to 100 MHz [9]. In Gouin O’Shaughnessey, P. ’s research, induction welding was performed at a frequency of 268 kHz [15]. The frequency used in the previous research was 750 kHz to 1 MHz. In this study, induction welding was performed at 100 kHz. When the magnetite content was twice or more than the thermoplastic resin, induction heating was possible even at a low frequency.

This study introduced a new Fe${}_{3}$O${}_{4}$ NP incorporated thermoplastic susceptor that was lightweight and flexible and easily used in induction welding. This susceptor is expected to lead to new applications in the field of thermoplastic composites.

## Figures and Tables

**Figure 1 materials-13-00318-f001:**

Fabrication process of the susceptor.

**Figure 2 materials-13-00318-f002:**
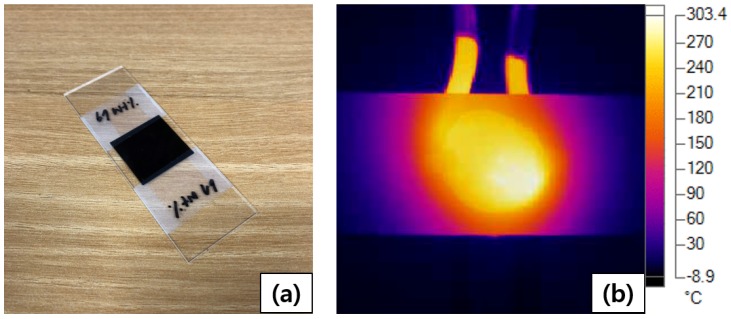
Heating temperature measurement using an infrared camera: (**a**) specimen and (**b**) measured image.

**Figure 3 materials-13-00318-f003:**
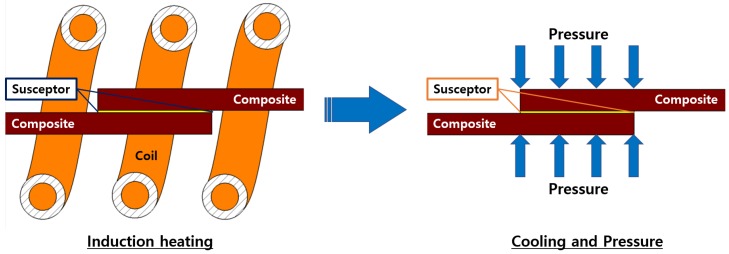
Induction welding process.

**Figure 4 materials-13-00318-f004:**
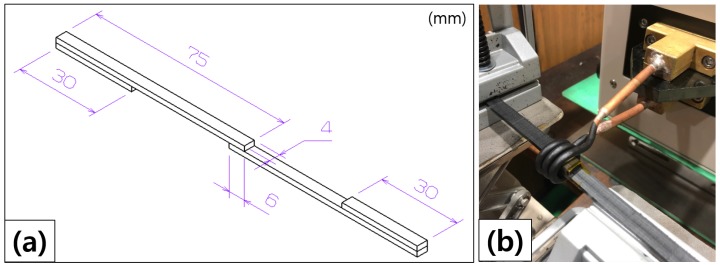
Induction welding of thermoplastic composites: (**a**) drawing of the specimen and (**b**) specimen and experimental setup.

**Figure 5 materials-13-00318-f005:**
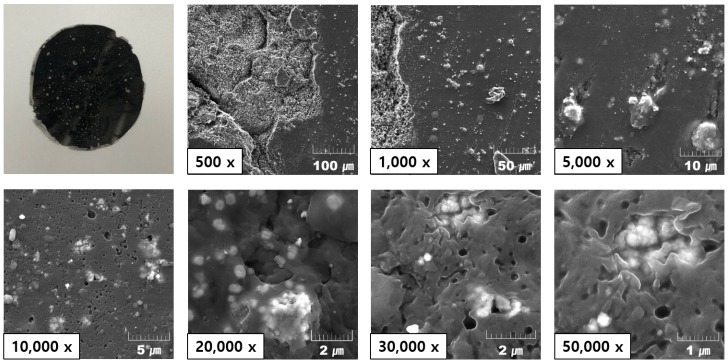
Scanning electron microscopy (SEM) image of the as-fabricated PA6/Fe${}_{3}$O${}_{4}$ (75 wt%) susceptor.

**Figure 6 materials-13-00318-f006:**
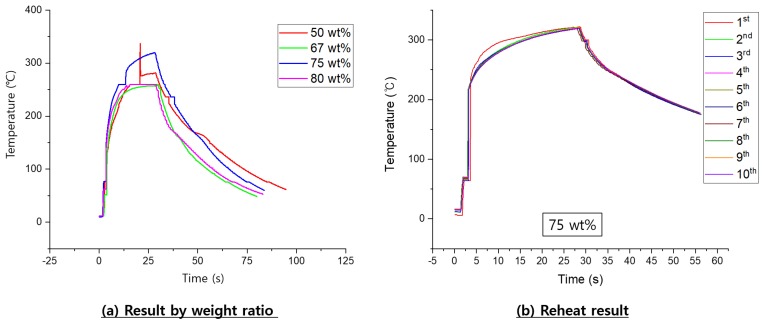
Induction heating results: (**a**) by Fe${}_{3}$O${}_{4}$ weight percentage and (**b**) during 10 iterations of heating.

**Figure 7 materials-13-00318-f007:**
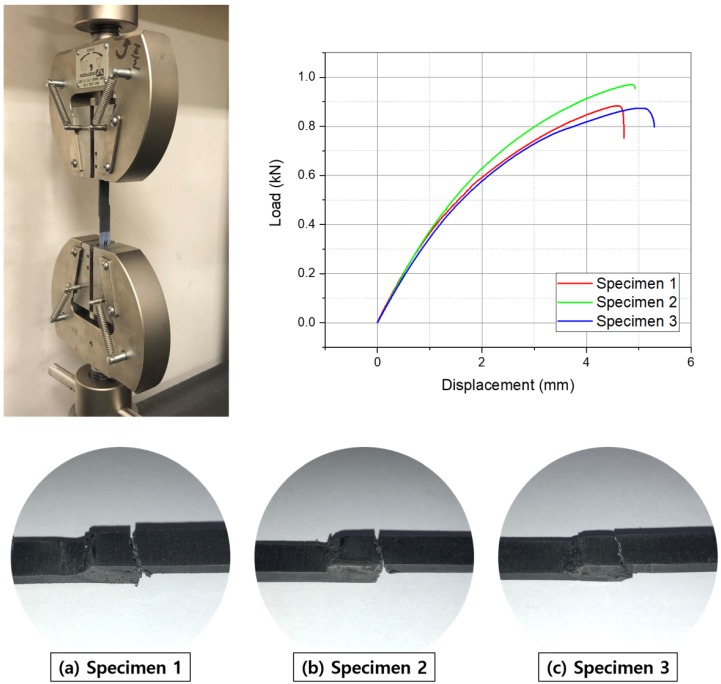
Adhesion failure after tensile tests: (**a**) Weld 1 after 883.8 N, (**b**) Weld 2, 816.7 N, and (**c**) Weld 3, 873.9 N.

**Figure 8 materials-13-00318-f008:**
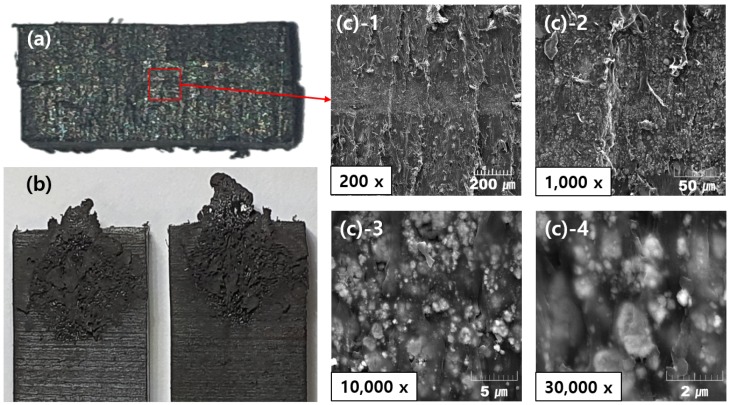
Cross-sectional SEM images of the weld joint at magnifications of 200–30,000: (a) Cross-section of specimen of the induction welding, (b) The appearance of the welded section, (c) Cross-section of induction welded specimen with magnification.

**Table 1 materials-13-00318-t001:** Powder type and working temperature of thermoplastic(TP) resin.

TP Resin	Powder Type	Working Temperature (${}^{\circ }C$)
PA6	O	180
PEEK	O	250
PPS	X	170

**Table 2 materials-13-00318-t002:** Masses of magnetite (Fe${}_{3}$O${}_{4}$) and polyamide 6 (PA6) used to form different weight ratios.

Material	50 wt%	67 wt%	75 wt%	80 wt%
PA6	100 g	100 g	100 g	100 g
Fe${}_{3}$O${}_{4}$	100 g	200 g	300 g	400 g

**Table 3 materials-13-00318-t003:** Tensile test results.

Case (Load)	Specimen 1 (883.8 N)	Specimen 2 (816.7 N)	Specimen 3 (873.9 N)	Deviation
**Lap shear strength**	36.8 MPa	34.0 MPa	36.4 MPa	29.51 MPa

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
