# Peer review of "A Study of Films Incorporating Magnetite Nanoparticles as Susceptors for Induction Welding of Carbon Fiber Reinforced Thermoplastic"

_materials, 2020, doi:10.3390/ma13020318_

Round 1

Reviewer 1 Report

This paper presents the development of a susceptor for induction welding of thermoplastic composites. Unfortunately, the susceptor that is developed is shown to lead to welded joints of poor mechanical performance with very low loads at failure. In addition, the paper contains important scientific and technical flaws. In particular, the experimental methodology is flawed and the results are not accurate nor reliable. For example, the authors report a LSS of around 33.8 MPa with load at failure of 810 N. This is incorrect as a load of 810 N corresponds to a LSS of 1.3 MPa (not 33.8 MPa). It is also not clear why magnetic particles are used here for a power supply capable of a maximum frequency of 400 kHz? To heat magnetic particles by hysterisis, frequencies in the range of 1-2 MHz are needed. In this case, the magnetic properties of the particles would not produce any heat and the heat generated is due to eddy currents only. Therefore other non-magnetic particles should have been used. I regret to say that this paper is not suitable for publication in a scientific journal.

The following list of comments gives more examples of incorrect reporting.

Introduction, first sentence: The authors state as the first sentente of the paper: "Thermosetting composites have commonly been used for these applications..." What applications are they talking about?

Introduction, line 24: Mechanical fastening is not used to bond composites but to assemble them.

Introduction, line 27: What is meant by mechanical bonding? Do the authors mean adhesive bonding?

Introduction, third paragraph: The authors should discuss that carbon fibre composites are usually welded without any susceptor.

Introduction, line 38: What is meant by adhesion failure within the resin. Please refer to the standard failure modes as defined in the literature.

Introduction, line 39: Please quantify the increase in weight that is claimed to come from the use of a stainless steel mesh relative to another susceptor or even the absence of susceptor

Introduction, last paragraph: Authors talk about susceptors and then heating elements. Please explain the difference between the two terms.

Introduction: The authors must discuss the different heating mechanisms of induction heating. It is not clear in this paper if heating is produced by eddy currents and Joule effect or by hysterisis heating.

Experiment

Please show micrographs of the cross-section of the welded samples. This should be added to the paper in an additional section. With such a high loading of particles, I doubt that the polymer can properly diffuse through the weld interface. This would explain why such a poor mechanical performance was obtained.

Section 2.2: What is meant by "degree of cooling"

Section 2.3 : Please give the dimensions of the adherents and welded overlap width and length.

Swction 2.3: It is well well known that the LSS depends not only on the welded polymer but on the fibre content, fibre nature, stacking sequence and thickness of the adherents. Please provide detail about the composite adherents.

Section 2.3: Please describe the induction welding process in more details. What is the appliaed current to teh coil? What is the coupling distance? What is the frequency?

Section 2.3 : What temperature is reached inside the weld after 30 sand how uniform is this temperature over the weld interface?

Section 2.4: The failure loads obtained are way too low for the load cell capacity (100 kN). Load cells are not accurate below 5-10% of their maximum capacity. In this case, a smaller load case must be used to test the welded joints to failure.

Results and Discussion

Section 3.1: What is meant by "excellent efficiency" Please explain how the efficiency is assessed.

Section 3.1: At what location is the temperature measured? How can IR be used to get the temperature of the susceptors inside the weld?

Author Response

Q 1. the experimental methodology is flawed and the results are not accurate nor reliable. For example, the authors report a LSS of around 33.8 MPa with load at failure of 810 N. This is incorrect as a load of 810 N corresponds to a LSS of 1.3 MPa (not 33.8 MPa).

Answer)

The width of the welded area can be seen in Figure 4. I also added content to the line 127.

Q 2. It is also not clear why magnetic particles are used here for a power supply capable of a maximum frequency of 400 kHz? To heat magnetic particles by hysterisis, frequencies in the range of 1-2 MHz are needed. In this case, the magnetic properties of the particles would not produce any heat and the heat generated is due to eddy currents only.

Answer)

I only got results with experiments. The article does not contain false content.

My experimental conditions can be seen in Figure 4 (c). The frequency is about 100 kHz.

I didn't know if frequency was an important point until you told me. Thank you very much.
In the next study, I will consider the frequency part.

Q 3. Introduction, first sentence: The authors state as the first sentente of the paper: "Thermosetting composites have commonly been used for these applications..." What applications are they talking about?

Answer) Added content. line 18-20

Many airliner structures, as well as automobile frames, are made from composites, because incorporation of composites offers the advantages of shorter processing times and reduced weight of the fabricated structure. Thermosetting composites have commonly.....

Q 4. Introduction, line 24: Mechanical fastening is not used to bond composites but to assemble them.

Answer) Added and modified content, line 35-46

Q 5. Introduction, line 27: What is meant by mechanical bonding? Do the authors mean adhesive bonding?

Answer)

This study does not use adhesives.
It use susceptors for induction welding.

Q 6. Introduction, third paragraph: The authors should discuss that carbon fibre composites are usually welded without any susceptor.

Answer) Commonly used welding processes are referred to as resistance welding and ultrasonic welding. line 48

Q 7. Introduction, line 38: What is meant by adhesion failure within the resin. Please refer to the standard failure modes as defined in the literature.

Answer) It is about the failure of welding of the metal mesh used previously.

Q 8. Introduction, line 39: Please quantify the increase in weight that is claimed to come from the use of a stainless steel mesh relative to another susceptor or even the absence of susceptor

Answer) I'm sorry, I did not find any weight gain data due to metal mesh.

My estimate is that the weight gain will be very small. Compared to my research, there is such a difference point.

Q 9. Introduction, last paragraph: Authors talk about susceptors and then heating elements. Please explain the difference between the two terms.

Answer) edited content, line 63-74

Q 10. Introduction: The authors must discuss the different heating mechanisms of induction heating. It is not clear in this paper if heating is produced by eddy currents and Joule effect or by hysterisis heating.

Answer) It is heated by the eddy current. line 51

Experiment

Q 11. Please show micrographs of the cross-section of the welded samples. This should be added to the paper in an additional section.

Answer) Welded area sections are included in Figure 8 (b).

Q 12. Section 2.2: What is meant by "degree of cooling"

Answer) Cooling means that the molten contact surface becomes solidification.

Q 13. Section 2.3 : Please give the dimensions of the adherents and welded overlap width and length.

Answer) You can see the dimensions of the welded face in Figure 4 (a).

Q 14. Swction 2.3: It is well well known that the LSS depends not only on the welded polymer but on the fibre content, fibre nature, stacking sequence and thickness of the adherents. Please provide detail about the composite adherents.

Answer) Information on fiber-reinforced plastics used in Section 2.1 can be found. It is not a laminated composite. Engineering plastics were used.

Q 15. Section 2.3: Please describe the induction welding process in more details. What is the appliaed current to teh coil? What is the coupling distance? What is the frequency?

Answer) AC current is applied to the induction heating coil. line 119

Sorry, I didn't understand what is the coupling distance.

The frequency of 100 kHz. line 118

Q 16. Section 2.3 : What temperature is reached inside the weld after 30 sand how uniform is this temperature over the weld interface?

Answer) Thermal imaging cameras cannot measure the internal temperature. Therefore, the temperature is unknown.

Q 17. Section 2.4: The failure loads obtained are way too low for the load cell capacity (100 kN). Load cells are not accurate below 5-10% of their maximum capacity. In this case, a smaller load case must be used to test the welded joints to failure.

Answer) Editing paper, a tensile test was performed with a capacity of 5 kN. line 124

Results and Discussion

Q 18. Section 3.1: What is meant by "excellent efficiency" Please explain how the efficiency is assessed.

Answer) Changed the phrase 'excellent efficiency' to 'highest temperature'.line 141

Q 19. Section 3.1: At what location is the temperature measured? How can IR be used to get the temperature of the susceptors inside the weld?

Answer) Welded surfaces can not measure temperature. Therefore, no temperature was measured during induction welding.

Reviewer 2 Report

The paper presents an interesting experimental work on the development of a PA6 film filled with magnetite powder to be applied as a susceptor in the induction welding of thermoplastic composites. The topic is hot and the work deserves publication if there some issues that need to be revised.

Introduction: should be improved with a comprehensive state of the art on the technologies for welding thermoplastic composites, including ultrasonic welding and resistance welding. Some recent refrences should be added: see for example Composites: Part A 82 (2016) 119, Adv Eng Mater 2017;19:11 and others. The state of the art on the use of magnetite in susceptors for induction welding should be added. See for example: Composites Research, 30, 2017, 181; Figure 2-a can be deleted and the process parameters can be added in the text in the Experimental section. The same for Fig 4c. The relative parameters should be added in the text. Figure 2b and c should be enlarges. In Figure 2b the different parts should be indicated with a number and explained in the text. A scale bar with the temperatures should be inserted in Fig 2c. Page 3: how is the pressure applied to the joining during the welding process? Fig 6b: the magnetite content should be specified. Since the maximum temperature is around 250 °C, Fig 6c does not refers to the 75 wt% of magnetite as written in the text. The authors should add in Fig 6 the plot after reheating concerning the 75% magnetite content. Page 4: the heating at 340 °C could be critical for an initial degradation of PA6. The authors should explain this point Figure 5: add a picture with the susceptors at the 4 weight content and explain the different morphologies, if there are. Page 6: the failure deserves a deeper analysis. Please add more comment on the failure type im the text. Although the authors declare that the English has been checked by two professional editors, the English form and style still requires some modifications. Some examples:

-Page 1 rows 18-19: rewrite the sentence that is not clear.

-Page 1 row 28: delete the sentence. What does "chemically inert thermoplastic " means?

-Page 2 row 43: change "fabricating" with "the use of "

-Page 4 row 100: change "the end of the prepared" with "obtained"

-Page 7 rows 141-145: the period is very unclear. Please change it carefully!

Author Response

Thank you so much for evaluating my research.

Introduction: should be improved with a comprehensive state of the art on the technologies for welding thermoplastic composites, including ultrasonic welding and resistance welding. Some recent refrences should be added: see for example Composites: Part A 82 (2016) 119, Adv Eng Mater 2017;19:11 and others. The state of the art on the use of magnetite in susceptors for induction welding should be added. See for example: Composites Research, 30, 2017, 181;

Answer) I'm sorry, I didn't understand. I searched for 'Composites: Part A 82 (2016) 119, Adv Eng Mater 2017; 19: 11','Adv Eng Mater 2017; 19:11', 'Composites Research, 30, 2017, 181' but the search did not any results.

Figure 2-a can be deleted and the process parameters can be added in the text in the Experimental section. The same for Fig 4c. The relative parameters should be added in the text. Figure 2b and c should be enlarges. In Figure 2b the different parts should be indicated with a number and explained in the text. A scale bar with the temperatures should be inserted in Fig 2c.

Answer) I corrected Figure 2. Added picture of the specimen. And i added a scale bar.

Page 3: how is the pressure applied to the joining during the welding process?

Answer) The Load is applied by applying the weight pendulum of 5 kg.

Fig 6b: the magnetite content should be specified. Since the maximum temperature is around 250 °C, Fig 6c does not refers to the 75 wt% of magnetite as written in the text. The authors should add in Fig 6 the plot after reheating concerning the 75% magnetite content.

Answer) I corrected Figure 6 b

Page 4: the heating at 340 °C could be critical for an initial degradation of PA6.

Answer) You are right. But only the susceptor is the temperature generated when 30 seconds of heating. When doing induction welding, The susceptors induction heating behavior is different. Therefore, I didn't mention it.

The authors should explain this point Figure 5: add a picture with the susceptors at the 4 weight content and explain the different morphologies, if there are.

Answer) The shape of the susceptor is almost identical. Therefore, 75 wt% was typically represented.

Page 6: the failure deserves a deeper analysis. Please add more comment on the failure type im the text.

Answer) The failed experiment was deleted. Successful experiments were further analyzed.

-Page 1 rows 18-19: rewrite the sentence that is not clear

Answer) Many airliner structures, as well as automobile frames, are made from composites, because incorporation of composites offers the advantages of shorter processing times and reduced weight of the fabricated structure. Thermosetting composites have commonly been...

-Page 1 row 28: delete the sentence. What does "chemically inert thermoplastic " means?

Answer) Yes, I deleted it.

-Page 2 row 43: change "fabricating" with "the use of "

Answer) line 60-61, Please, read it again.

Other researchers have investigated the feasibility of 60
fabricating a thin-film susceptor from a mix of powdered ferromagnetic material and thermoplastic resin, and several studies have presented new susceptor configurations.

-Page 4 row 100: change "the end of the prepared" with "obtained"

Answer) Yes, I corrected it. line 134.

-Page 7 rows 141-145: the period is very unclear. Please change it carefully!

Answer) I completely changed the content. line 169-173.

In this study, we fabricated thin-film susceptors for induction welding from a thermoplastic composite made of PA6 resin and Fe3O4 NP additive. The good weight percentage for susceptor fabrication was identified as 75 wt% Fe3O4. Induction welding was performed using this susceptor, and the weld strength and quality were verified by tensile tests and SEM analysis of the weld joint cross-section. The results are summarized below.

Reviewer 3 Report

Please find enclosed the manuscript with questions and comments/suggestions. 

Generally, I am missing the discussion on what the described work is bringing to the state of the art. This could be described more thoroughly in the Introduction. Furthermore, the discussion of the Results should be deepened, also in relation to previous publications or literature references. Also, the composite materials joined should be described more thoroughly. 

Good luck!

Kind regards

Author Response

Thank you for your evaluation of my research.

I corrected your point. Please check the attached file and try again.

Q. Nonetheless, it would be fair to detail about the advantages and disadvantages of both.

Aswer) You can see the line 18-34

Q. The disadvantages of mechanical fastening are not related only to thermoplastics. Furthermore, this phrase mixes adhesive and mechanical fastening without sufficient details. Please update.

Answer) You can see the line 35-46

Q. The last sentence or paragraph of the Introduction should specify what kind of contribution this work will bring. This is not clear at the moment. 

Answer) You can see the line 63-74

Q. One section describing the properties of the base material composites to be welded is missing.

Answer) You can see the line 76-85

Q. please use a synonym which describes better the production of the susceptor

Answer) You can see the line 86.

Q. On which basis where the welding parameters selected/determined?

Answer) Induction welding parameters were determined by trial and error.

Q. How many replicates were tested? 

Answer) There is a lack of material from which the specimen can be made. Thus, the tensile test was performed five times. In editing the paper, it was pointed out that the load and head speed of the tensile tester were high. So I did a tensile test again. Of the five tensile test results, I included three test results.

Q. were there any other, lower speeds tested as well?

Answer) You can see the line 134

Tensile tests were conducted with a load of 5 kN and a head speed of 1 mm / min.

Q. How many samples per susceptor were tested? 

Answer) I did three tests per susceptor. The test results were almost identical.

Q. Please mention the wt%. I assume it is 75 for this sample. 

Answer) You can see the figure 5

Q. This part requires more discussion. Please describe the failure mechanism and the reasons for it. Furthermore, what is the learning effect of this test? 

Answer) You can see line 159-167

The failed test results were deleted. Instead, further discussion on induction welding results was added.

Q. Please add a column with the standard deviation 

Answer) You can see the table 3

Q. It is not clear why 75% was identified as optimal. Please detail.

Answer) You can see line 170

The word 'optimal' is wrong. Corrected to the expression 'selected'. Induction heating temperature was shown to be high, and the content which stable heating behavior was selected.

Q. Which is the difference to other susceptors from literature / other works / state of the art? In value and behavior. 

Answer) You can see line 198-199

Reviewer 4 Report

A new susceptor comprised of Fe3O4 nanoparticles and thermoplastic PA6 is fabricated for the induction welding of PA6/CF in this study. The first revision has improved the quality of the manuscript. However, there are still several comments that need to be addressed:

1. Although the manuscript can be followed, the language should be further improved to satisfy the standard of scientific paper.

2. Figs. 5 and 8 need scale bars.

3. The Conclusions should extract the main findings and innovations of the investigation. In the manuscript, many contents in the Conclusions are not mentioned in the former sections, such as “PA6 and Fe3O4 must be mixed….. , resulting in a poor susceptor”. Most of the items in the Conclusions are the discussion and  introduction of the experiment processes. The Conclusions must be revised to delete the useless contents and become more impact.

Author Response

Q1. Although the manuscript can be followed, the language should be further improved to satisfy the standard of scientific paper.

A. We have tried to improve the language. The English in this document has been checked by at least two professional editors, both native speakers of English.

Q2. Figs. 5 and 8 need scale bars.

A. Paper has been modified.

Q3. The Conclusions should extract the main findings and innovations of the investigation. In the manuscript, many contents in the Conclusions are not mentioned in the former sections, such as “PA6 and Fe3O4 must be mixed….. , resulting in a poor susceptor”. Most of the items in the Conclusions are the discussion and  introduction of the experiment processes. The Conclusions must be revised to delete the useless contents and become more impact.

A. Paper has been modified.

Round 2

Reviewer 1 Report

Unfortunately, the comments provided in the first version of the paper were not properly addressed.

How is it possible that the authors did not consider (did not even know) the importance of the magnetic field frequency for induction heating based on hysteresis?

The experimental results, including the reported lap shear strengths are invalid as the welded samples are extremely small compared to what is reported in the literature. Contrarily to what the authors claim, ASTM standard D5878 was not followed as the sample geometry is different from what the standard prescribes.

Most of the comments provided previously remain unaddressed and I recommend rejecting this paper.

Author Response

Thank you for your feedback.

Q. How is it possible that the authors did not consider (did not even know) the importance of the magnetic field frequency for induction heating based on hysteresis?

Answer) This paper wanted to show the success of induction welding.

I experimented with our's equipment and found out that induction heating is possible. So I didn't study the electromagnetic properties (Hysteresis, frequency) of magnetite. I found that hysteresis and frequency are important with reviewer feedback. 

Q. The experimental results, including the reported lap shear strengths are invalid as the welded samples are extremely small compared to what is reported in the literature. Contrarily to what the authors claim, ASTM standard D5878 was not followed as the sample geometry is different from what the standard prescribes.

Answer) Our induction heating equipment cannot be tested in the regular size according to ASTM 5868. Thus, the specimens in this study were made smaller than ASTM 5868. 

The LSS stress of this study was higher than 22.2 ~ 31.3 MPa mentioned in the previous study. I was judged that induction welding was successful because of excellent results.

I attached video. Please, confirm the induction welding process of this paper by video.

Reviewer 2 Report

The paper has still some flaws. Since the modifications are not highlighted in the text, it is not straightforward to idetify them. I suggest to the authors to highlight in yellow the modifications in the manuscript, as requested in the Guide for the Authors and to resubmit the Revisions.

The suggested references can be found on Google Scholar:

- "Modeling of continuous ultrasonic impregnation and consolidation of thermoplastic matrix composites." Composites Part A: Applied Science and Manufacturing 82 (2016): 119-129;

- "Novel Heating Elements for Induction Welding of Carbon Fiber/Polyphenylene Sulfide Thermoplastic Composites". Adv. Eng. Mater., 19: 1700294. doi:10.1002/adem.201700294

- " Comparison of Heating Behavior of Various Susceptor-embedded Thermoplastic Polyurethane Adhesive Films via Induction Heating", Composites Research Vol. 30, No. 3, 181-187 (2017)

As concerning my comments "Page 4: the heating at 340 °C could be critical for an initial degradation of PA6", the author's answer "Answer) You are right. But only the susceptor is the temperature generated when 30 seconds of heating. When doing induction welding, The susceptors induction heating behavior is different. Therefore, I didn't mention it" is not convincing. I suggest to the authors to add a pertinent comment also in the manuscript since this issue cannot be neglected in the analysis.

Author Response

Thank you for your feedback.

The suggested references can be found on Google Scholar:

- "Modeling of continuous ultrasonic impregnation and consolidation of thermoplastic matrix composites." Composites Part A: Applied Science and Manufacturing 82 (2016): 119-129;

- "Novel Heating Elements for Induction Welding of Carbon Fiber/Polyphenylene Sulfide Thermoplastic Composites". Adv. Eng. Mater., 19: 1700294. doi:10.1002/adem.201700294

Answer) I added references.

 " Comparison of Heating Behavior of Various Susceptor-embedded Thermoplastic Polyurethane Adhesive Films via Induction Heating", Composites Research Vol. 30, No. 3, 181-187 (2017)

This document is already referenced.

Q. As concerning my comments "Page 4: the heating at 340 °C could be critical for an initial degradation of PA6", the author's answer "Answer) You are right. But only the susceptor is the temperature generated when 30 seconds of heating. When doing induction welding, The susceptors induction heating behavior is different. Therefore, I didn't mention it" is not convincing. I suggest to the authors to add a pertinent comment also in the manuscript since this issue cannot be neglected in the analysis.

Answer) Conclusion section, line 191-194, 

I added content.

Reviewer 3 Report

Thank you for addressing my comments and questions and for the updates. 

I included a few remarks that could improve the work's contribution further. 

Good luck and kind regards

Author Response

Thank you for your feedback

Q. Is there any supposition that can be made related to how the Fe3O4 behaves within the process (temperature, material properties, loads, etc), that would lead to this well mixing and uniformity, compared to other susceptors? It would add up to the discussion and the value of the work to reference this to other works, from this point of view. 

Answer) line 202-206, I added content. The main difference from the preceding susceptor is induction welding with the same material as the adherend. In addition, nanoparticles are included, so induction heating is possible even after induction welding.

Reviewer 4 Report

The manuscript has been improved after modification. It is recommended to be published.

Author Response

Thank you for reviewed the paper.

Round 3

Reviewer 1 Report

Carbon fibre, even short carbon fibre, can be heated by induction. What is the relative contribution of the susceptor and fibre to the heat generation?

Please give more details about the susceptor fabrication. How were the Fe3O4 particles mixed in the polymer? Using an extruder?

What is the temperature at the weld interface during induction heating? Can the author provide a curve of the temperature as function of time as is usually done during induction welding?

Please give more information about the heating behaviour of the joints (and not just the susceptor) and indicate the temperature of the weld interface at the time when the mass of 5 kg is applied on the specimens. Please add a figure similar to fig 6 with the heating of the joints (and not just the susceptors)

Section 2.4 still incorrectly indicates that ASTM Standard D5868 was followed.

Please discuss the effect of a small weld surface area (6 mm X 4mm  as opposed to standard 25.4 mm X 25.4 mm) on the state of stress developed during the lap shear test and the apparent lap shear strength.

As stated in the introduction, the gola of this study was to "improve the induction welding performance of thermoplastic composites by fabricating new susceptors". What is meant by "induction welding performance"? Did the authors mean "mechanical performance of welded joints"? If so, please discuss the fact that the manufactured joints are very very small and do not represent any improvement in mechanical performance. Also, the manufactured susceptors are not new as highlighted by the literature review.

Author Response

Q. Carbon fibre, even short carbon fibre, can be heated by induction. What is the relative contribution of the susceptor and fibre to the heat generation?

A. you're right. At high frequencies, induction heating is possible only with carbon fiber. However, this study does not intend to heat the entire composite. It is a method of joining / welding by partial induction heating. Induction heating behavior is a method that can be bonded / welded with much higher magnetite than carbon fiber.

Q. Please give more details about the susceptor fabrication. How were the Fe3O4 particles mixed in the polymer? Using an extruder?

A. We didn't use an extruder. Please, check the attached video.

What is the temperature at the weld interface during induction heating? Can the author provide a curve of the temperature as function of time as is usually done during induction welding?

A. Temperature measurement is difficult by magnetic field. In the following paper, Temperature measurements are made using fiber-optic thermometers.

Q. Please give more information about the heating behaviour of the joints (and not just the susceptor) and indicate the temperature of the weld interface at the time when the mass of 5 kg is applied on the specimens. Please add a figure similar to fig 6 with the heating of the joints (and not just the susceptors)

A. Temperature measurement should be measured using a fiber-optic thermometer. But in paper, it didn't proceed. We will add content to the next paper.

Q. Section 2.4 still incorrectly indicates that ASTM Standard D5868 was followed.

A. Paper has been modified.

Q. Please discuss the effect of a small weld surface area (6 mm X 4mm  as opposed to standard 25.4 mm X 25.4 mm) on the state of stress developed during the lap shear test and the apparent lap shear strength.

A. The standard weld joint area is 25.4 mm X 25.4 mm. The size of the induction heating coil in this study is 15 mm in inner diameter. Therefore, the size of the specimen was set small. In this study, LSS results are equivalent to those generated from standard specimens.

As stated in the introduction, the gola of this study was to "improve the induction welding performance of thermoplastic composites by fabricating new susceptors". What is meant by "induction welding performance"? Did the authors mean "mechanical performance of welded joints"? If so, please discuss the fact that the manufactured joints are very very small and do not represent any improvement in mechanical performance. Also, the manufactured susceptors are not new as highlighted by the literature review.

A. Paper has been modified.

2 page, 73 line, "improve" => "verify"

The meaning of "welding performance" can be found in lines 71-73.

To date, research on induction heating behavior has been conducted.

This paper verified induction heating behavior and bonding / welding performance. So I described it as a new susceptor.

Reviewer 2 Report

I recognize the efforts of the authors in improving the paper.  I encourage them to do a further little effort to improve some issue. I recommend the publication under minor revisions.

My comments are below:

1. The state of the art on the use of magnetite or other micron-sized (or smaller) heating particles in susceptors for induction welding is too qualitative.

2. The improvements or drawbacks are not analyzed. The authors should review better the literature in order to highlight the originality of the work and the achieved improvement.

3. In Figure 2b the different parts should be indicated with a number and explained in the text. Figure 4c can be deleted and the process parameters should be added in the text in the Experimental section.

4. Page 4: the heating at 340 °C could be critical for an initial degradation of PA6. The authors should explain this point.

5. The frequency of 100 kHz is too low. Usually in the literature higher frequencies are used. The authors should add a comment on this issue.

Author Response

 Q1. The state of the art on the use of magnetite or other micron-sized (or smaller) heating particles in susceptors for induction welding is too qualitative.

A. I think that method of using very small size magnetite is not qualitative. Many researchers are working on very small particle sizes of magnetite.

    I have suggested a magnetite content suitable for induction heating. And the possibility of induction welding was examined and the induction welding was successfully performed.

    I added more references that are similar to my research. [References (9), (10), (11), (20)]

Q2. The improvements or drawbacks are not analyzed. The authors should review better the literature in order to highlight the originality of the work and the achieved improvement.

A. References have been added to the introduction of the paper. (line 49)

    Section Refereces 9, 10, 11, 20 (line 234 ~ 240, line 258 ~ 260)

Q3. In Figure 2b the different parts should be indicated with a number and explained in the text. Figure 4c can be deleted and the process parameters should be added in the text in the Experimental section.

A. Modified Figure 2 and Figure 4. (removed Figure 2(b), Figure 4(c)),

    Paper has been modified => "The induction heating behavior was examined using an infrared camera(Ti450PRO, FLUKE, U.S.A., -10 C – 1500 C)."  (line 96, 97)

    "The power output for the experiments was 3.4 kW at a frequency of 100 kHz. And ouput current is 45 A for the induction welding experiments." (line 119 ~ 121)

Q4. Page 4: the heating at 340 °C could be critical for an initial degradation of PA6. The authors should explain this point.

A. Ppaer has been modified. =>  "In particular, the induction heating of the susceptor should be at least 300 C. When the susceptor is more than 300 C, it is heat transfer enough to be welded to the susceptor and the joint surface." (line 116 ~ 118)

    "When only the susceptor was induction heated for 30s, it was heated to 340 C. Because of the temperature is transfered to the joint surface during induction welding." (line 187 ~188)

Q5. The frequency of 100 kHz is too low. Usually in the literature higher frequencies are used. The authors should add a comment on this issue.

A. In reference (9) induction heating is used from 60 Hz to 100 MHz. Reference (20) uses a frequency of 268 kHz.
